



# Using ICESat-2 and Operation IceBridge altimetry for supraglacial lake depth retrievals

Zachary Fair[1], Mark Flanner[1], Kelly Brunt[2], Helen Amanda Fricker[3], and Alex Gardner[4]

[1]University of Michigan, Ann Arbor, MI
[2]NASA Goddard Space Flight Center, Greenbelt, MD
[3]Scripps Institution of Oceanography, San Diego, CA
[4]NASA Jet Propulsion Laboratory, Pasadena, CA

**Correspondence:** Zachary Fair (zhfair@umich.edu)

**Abstract.** Supraglacial lakes and melt ponds occur in the ablation zones of Antarctica and Greenland during the summer months. Detection of lake extent, depth, and temporal evolution is important for understanding glacier dynamics, but passive remote sensing techniques have inherent uncertainties associated with depth retrievals, and observations from the original ICESat mission experienced high absorption in water. In this study, we use laser altimetry data from the Ice, Cloud, and

land Elevation Satellite-2 (ICESat-2) over the Antarctic and Greenland ablation zones and the Airborne Topographic Mapper (ATM) for Hiawatha Glacier (Greenland) to demonstrate retrievals of supraglacial lake depth. Using an algorithm to separate lake surfaces and beds, we present case studies for 12 supraglacial lakes with the ATM lidar and 12 lakes with ICESat-2. Both lidars detect bottom returns for lake beds as deep as 7 m. Uncertainties for these retrievals are 0.05-0.20 m for ATM and 0.12-0.80 m for ICESat-2, with the highest uncertainties observed for lakes deeper than 4 m. Using ICESat-2 confidence

classifications of detected photons, we found that high-confidence photons are often insufficient to fully profile lakes, so lower confidence and buffer photons are recommended for improved retrievals. Despite issues in photon classification, the altimeter results are promising, and we expect them to serve as a benchmark for future studies of surface meltwater depths.

## 1 Introduction

The ice sheets of Antarctica and Greenland modulate rates of sea level rise, contributing $14.0 \pm 2.0$ mm (Antarctica) and $13.7 \pm$

$1.1$ mm (Greenland) since 1979 (Mouginot et al., 2019; Rignot et al., 2019). Current trends indicate greater melt in the coming decades, leading to both ice sheets becoming dominant contributors to sea level rise (Vaughan et al., 2013). Meltwater plays vital roles in ice sheet evolution (e.g., van den Broeke et al., 2016), including accumulation on ice sheets as supraglacial lakes, many of which are several meters deep (Echelmeyer et al., 1991). These lakes exhibit a lower albedo than the surrounding ice, allowing them to absorb more incoming solar radiation and melt ice more efficiently, thus generating a positive feedback (Curry

et al., 1996). Supraglacial lakes are significant reservoirs of latent heat (Humphrey et al., 2012), and their spectral emissivity in the IR spectrum also differs from bare ice (Chen et al., 2014; Huang et al., 2018), leading to potentially significant impacts on the surface energy balance of ice sheets.




A substantial portion of meltwater eventually drains into supraglacial streams or moulins (drainage channels), where it can flow to the ice bed (Banwell et al., 2012; Catania et al., 2008; Selmes et al., 2011). Meltwater penetration into the ice also leads to hydrofracture, a mechanism through which meltwater facilitates full ice fracture as a result of the stresses induced by the density contrast between liquid water and ice (Das et al., 2008). Meltwater injection to the bed can also modify basal

water pressures which in turn modify the resistance to ice flow and thus sliding velocity (Parizek and Alley, 2004; Zwally et al., 2002). Hydrofracture can lead to significant ice loss for outlet glaciers and ice shelves (Banwell et al., 2013). Current observations and modeling efforts indicate a propagation of supraglacial lakes farther inland as the climate warms (Howat et al., 2013; Leeson et al., 2015; Lüthje et al., 2006), raising further concerns for accelerated mass loss. For these reasons, knowledge of supraglacial lakes is important for our understanding of ice sheet evolution.

Previous studies developed techniques for detecting supraglacial lakes and retrieving depth, areal coverage, and volume. In-situ observations employed sonar and radiometers to approximate lake depth and albedo (Box and Ski, 2007; Tedesco and Steiner, 2011). However, the harsh conditions of Antarctica and Greenland, the transience of meltwater, and the sheer size of the ice sheets restrict the potential for extensive in-situ measurements, encouraging lake depth and areal coverage estimates from passive remote sensing data such as Landsat-8, MODIS, and Sentinel-2 A/B. Supraglacial water is darker than surrounding

ice in visible and IR bands, allowing the use of band ratios between blue and red reflectance (Stumpf et al., 2003). The normalized water difference index (NWDI) and dynamic thresholding techniques have also been considered for lake detection (Fitzpatrick et al., 2014; Liang et al., 2012; Moussavi et al., 2016; Williamson et al., 2017; Moussavi et al., 2020). Other methods implemented radiative transfer models (Georgiou et al., 2009) or positive degree day models (McMillan et al., 2007) to estimate lake albedo and meltwater volume, respectively. By comparing surface reflectance data of supraglacial water to that

of ice and optically deep water, empirical relationships have been derived to approximate lake depth (Philpot, 1989; Sneed and Hamilton, 2007).

Image-based empirical techniques rely on approximations of lake bed albedo and an attenuation parameter, both of which are subject to uncertainties from lake heterogeneity and cloud cover (Morassutti and Ledrew, 1996). Furthermore, Pope et al. (2016) found that band ratios were insensitive to lakes deeper than 5 m, leading to errors that may exceed 1 m. Parameter

fitting in the empirical equations requires supplementary depth retrievals, often from in-situ sources. More accurate methods for supraglacial lake detection are needed to improve image-based estimates.

In September 2018, the Ice, Cloud and land Elevation Satellite-2 (ICESat-2) with the primary objective of obtaining laser altimetry measurements of the polar regions (Abdalati et al., 2010; Markus et al., 2017; Neumann et al., 2019b). Observations using the Airborne Topographic Mapper (ATM) and Multiple Altimeter Beam Experimental Lidar (MABEL) indicated the

potential for shallow water profiling with laser altimetry (Brock et al., 2002; Brunt et al., 2016; Jasinski et al., 2016), and ICESat-2 applications were recently demonstrated by Ma et al. (2019) and Parrish et al. (2019). In this study, we identify test cases from ICESat-2 and ATM altimetry data and use these pilot cases to develop an algorithm for detecting supraglacial lakes and retrieving lake depth. The algorithm is designed as a semi-automatic method to find supraglacial lakes within select altimetry granules.



## 2 Data Description

### 2.1 ICESat-2

ICESat-2 is a polar orbiting satellite with an inclination of 92 degrees that carries the Advanced Topographic Laser Altimeter System (ATLAS), a 532 nm micro-pulse laser that is split into 6 distinct beams: GT1L/R, GT2L/R, and GT3L/R. The beams

are configured in pairs with a 90-meter separation between beams within a beam-pair and 3.3-kilometer between pairs. With an operational altitude of ~500 km and a 10 kHz pulse repetition rate, ICESat-2 records a unique laser pulse every 0.7 m along-track over a 91-day repeat cycle.

The ATLAS product used here is the ATL03 Global Geolocated Photon Data V002 (Neumann et al., 2019a), which consists of retrieved photons tagged with latitude, longitude, received time, and elevation. Each photon pulse also carries a classification

as either signal or background (noise). The differentiation between signal and background is performed using a statistical algorithm outlined by Neumann et al. (2019b). Signal photons are further classified by confidence level, such that photons labeled as "high confidence" are most likely to originate from the surface. Generally, cloudy or variable profiles exhibit "medium/low confidence" or noise photons, whereas low slope surfaces, such as water and ice sheets, result in more "high confidence" photons (Neumann et al., 2019b). In thin layers of water, high confidence photons are observed from both the water surface and

the underlying ice.

Of the six beams available, we concentrated on the central strong beam (GT2R) to increase the likelihood of detecting lake bottoms. Ground-based validation by Brunt et al. (2019b) indicates an accuracy of <5 cm in ATL03 photons over ice sheet interiors. Only high-confidence photons were considered initially, but photons of lower confidence were included for attenuated lake bottoms (see Sect. 5.2 for more details). The addition of medium, low, and "buffer" photons slightly decreases

measurement precision but gives better agreement with ground-based data (Brunt et al., 2019b).

### 2.2 Airborne Topographic Mapper

The Airborne Topographic Mapper is a 532 nm lidar flown as part of Operation Icebridge (OIB), a campaign designed to fill the gap in polar altimetry between ICESat and ICESat-2. The ATM lidar conically scans at 20 Hz, providing a 400 m swath width along-track (Brock et al., 2002; Krabill et al., 2002). The ATM Level-1B Elevation and Return Strength (ILATM1B)

product converts analog waveforms into a geolocated photon product to emulate ATLAS data (Studinger, 2013, updated 2018), though it lacks a statistical confidence definition. Despite this, Brunt et al. (2019a) found that ATM errors were -9.5 to 3.6 cm relative to ground-based measurements. Currently, the ATM results presented serve as a proof of concept for the lake detection algorithm.



## 3 Methods

### 3.1 Lake Detection

Supraglacial lake surfaces are much flatter than surrounding terrain. We thus performed topography checks with the expectations that (i) lake surfaces would be easily identifiable in photon histograms and (ii) lake beds may be found via statistical inference in the region of the lake surface. To simplify the identification of lake features, we separated them into two arrays: one for the surface and one for the bed. We refer to this technique as "lake surface-bed separation" (LSBS). For both lidars, the procedure for separation was identical, and is as follows (see Fig. 2 for a schematic view):

- We divided each data granule into discrete along-track windows to reduce the data volume to ~$10^4$-$10^5$ photons per window. If a supraglacial lake appeared on the edge of the window, the window size was adjusted to include the full observed water feature.

- Each data window was binned into elevation-based histograms. We assumed that the lake surface dominates the total bin count within each window of $10^5$ photons. Thus, we performed a flatness check by computing the standard deviation of high-confidence signal photons within the upper 85th-percentile of bin count. A flat water surface was defined for ATL03 data where $\sigma \leq 0.05$ m, or $\leq 0.002$ m for ILATM1B data. If data were within the appropriate flatness threshold, they were verified as a lake surface using Landsat-8 OLI imagery. This step was included to filter non-glacial features, such as ocean or fjords. If the satellite image(s) confirmed the presence of a lake, the data were assigned to a new array for the lake surface ($h_{sfc}$).

- The horizontal extent of where the above criteria were met served as a constraint for where the lake bottom data could be defined. Within these horizontal bounds, high-confidence photons were defined as a lake bottom if they satisfied the condition: $h_{sfc} - 1.8\,\sigma_{sfc} \leq h \leq h_{sfc} - 0.75\,\sigma_{sfc}$, where $\sigma_{sfc}$ is the standard deviation of lake surface photons. We set this constraint to reduce the impact of multiple scattering on depth estimates. If these conditions were met, then the data were placed in an array for the lake bottom, $h_{btm}$.

- Lake bed photons often are classified at a lower confidence (Sect. 5.2), necessitating the inclusion of lower confidence levels. Notably, Greenland supraglacial lakes observed by ICESat-2 featured this issue if they exceeded 3 m in depth. For these cases, the condition for $h_{btm}$ was revised as $h_{sfc} - \sigma_{sfc} \leq h \leq h_{sfc} - 0.5\,\sigma_{sfc}$ to filter background photons within the water column.

- A series of filters were applied to improve surface/bed estimates. For ICESat-2, lakes shallower than 1.3 m or smaller than 200 m in horizontal extent were considered too noisy or ill-defined for further analysis. To remove water bodies with deep bed returns (e.g., oceans or fjords) or with no bed returns, the algorithm counted the number of bed photons present for both lidars. If the number of bed photons was very small (100 or less), then the scene was marked as a false positive.





## 3.2 ATL03 Refinement

The above steps were sufficient to obtain lake profiles within the ATM data, but melt lake bottoms observed by ICESat-2 were significantly noisier as a consequence of higher background (noise) photon rates. After the initial LSBS procedure, we manually assessed bed estimates for each lake. For lakes that did not pass qualitative assessment, we adopted photon

5   refinement procedures initially used for the ATL06 surface-finding algorithm (Smith et al., 2019). In short, ATL03 photon aggregates within 40 m segments were used to estimate lake surfaces and beds with greater precision via least-squares linear fitting applied to the aggregates. These linear fits were used to approximate a window of acceptable surface or bed photons for every 20 m along-track. A more detailed description of the ATL06 algorithm is given in Smith et al. (2019).

The linear regression in ATL06 accounts for all photons (background or signal), and the technique performs a background-

10   corrected spread estimate to narrow the range for acceptable photons. Background photons are omitted from LSBS refinement, and low-/medium-confidence photons are only considered if the high-confidence photons are deemed insufficient. The ATL06 refinement process also repeats its acceptable photon filter until no photons are removed. In LSBS, the flatness of lake surfaces and relatively low photon density of the corresponding beds rendered iterating unnecessary. Finally, the condition for acceptable surface photons in ATL06 is given by:

$$|r - r_{med}| < 0.5H_w \tag{1}$$

Within a 40 m photon segment, $r$ is the residual of a photon relative to the linear regression, $r_{med}$ is the median residual, and $H_w$ is window height. The height of the window is taken as the maximum of the observed photon spread, the previous window height (if any), and 1 m, and photons within the window range are defined as the surface. The lake bed is then defined as photons not within the window and below the surface. In other terms, lake bed photons satisfy the conditions:

$$|r - r_{med}| > 0.5H_w, \quad h < h_{sfc} \tag{2}$$

As with the initial guess, the lake bottom was only defined within the horizontal bounds of the lake surface, and the improved guesses were assigned to $h_{sfc}$ and $h_{btm}$.

As a final adjustment to lake photons, we applied a refraction correction algorithm to account for spatial offsets from the change in media. The correction follows the methods utilized by Parrish et al. (2019) by approximating refractive biases as a

25   function of depth and beam elevation angle. The center strong beam for ICESat-2 is near-nadir, so the horizontal offset was determined to be small relative to the size of lakes (~3 cm for a lake 10 m deep). However, vertical offsets of 1 m or more were found for lakes ≥4 m in depth, necessitating the use of refraction correction.



### 3.3 Lake Depth and Extent Estimations

Once we obtained $h_{sfc}$ and $h_{btm}$, lake depth from the altimeter signal ($z_s$) was estimated using:

$$z_s = \overline{h}_{sfc} - \overline{h}_{btm} \tag{3}$$

where $\overline{h}_{sfc}$ and $\overline{h}_{btm}$ represent the moving mean of the surface elevation and the bottom elevation, respectively. The moving

mean was used to account for signal attenuation and scattering at the lake bottom, a problem most evident over the Amery Ice Shelf.

For deep or inhomogeneous lakes, water attenuation resulted in fewer signal photons observed at lake bottoms. To approximate lake depths in these situations, we fitted a 3rd-order polynomial to all lake profiles with bounds at the lake edges. Lake depths approximated in this manner were denoted as $z_p$. We compare $z_s$ and $z_p$ over lakes with well-defined bottoms, and show

that the two generally agree well, in Sect. 4.

To test the limits of the algorithm relative to lake size, we utilized the great-circle formula (ATM) or pre-defined along-track distance (ICESat-2) to approximate along-track extent $L$. We acknowledge the further potential for lake volume retrieval, but improved estimates of lake radius and shape through visible imagery are required. We leave the development of such an algorithm for a future study.

### 15  3.4  Case Study Locations

We present cases over the Amery Ice Shelf on 2 January, 2019 (ICESat-2 Track 0081; 68.271-73.798°S, 63.057-78.620°E), the western Greenland ablation zone for 17 June, 2019 (ICESat-2 Track 1222; 66.575-69.582°N, 48.284-49.239°W), and Hiawatha Glacier in 19 July, 2017 (ATM; 77.780-79.3119°N, 65.279-67.484°W) (Fig. 1). Comparisons between Landsat-8 imagery and ICESat-2/OIB flight tracks confirmed supraglacial lake overpasses for study. The tracks examined for this

analysis are listed in Table 1. In Spring 2019, an early onset of the Arctic melt season resulted in both ICESat-2 and Operation Icebridge surveying supraglacial lakes near Jakobshavan Isbræ in May. Data granules from ATM for this time period were not available for download at the time of publication, but we expect these observations to be useful for additional depth estimates and validation in a future study.

### 4  Results

We detected a total of 16 supraglacial lakes in the ATM data. Of those, four were omitted from further analysis due to poor signal return from the lake bed. The remaining lake profiles are shown in Fig. 3, with maximum depths of 0.98-7.38 m and extents of 180-730 m. The algorithm identifies lake surfaces with good accuracy, and lake bottoms are well-defined in all but the deepest lakes. Lake bottoms >8 m below the surface exhibit fewer signal returns, for the associated return signal is below the threshold required to be digitized (Martin et al., 2012). Average measured lake depth was estimated as 1.95 m (Table 2),

and lakes at this depth typically featured adequate bed returns. In deeper lakes, the polynomial estimate produced reasonable





guesses for the lake bed location, with the most effective fitting seen in lakes 3e, 3g, and 3h. With the polynomial-based depths, mean lake depth increased to 2.15 m, and the maximum modeled depth was 8.83 m.

The spread in ATM lake bed photons is low (Table 2, Column 7), with a maximum of 0.2 m for lake 3g. The largest uncertainties are observed for lakes deeper than 3 m, perhaps influenced by low photon returns and the conical scanning of the

5 lidar. Polynomial estimation errors are 0.41 m on average. Several depth errors are below this mean, but a strong standard error (1.03 m) in lake 3g, due to difficulties in capturing its steep bed slope, slightly skews the mean error. Excluding this value, the mean error among ATM polynomial estimates reduces to 0.35 m.

We examined an additional 12 supraglacial lakes with ICESat-2, eight in Greenland and four also explored by Magruder et al. (2019) and Fricker et al. (in prep.) on the Amery Ice Shelf. The refined algorithm captured these lake surfaces and beds

with reasonable success, as seen in Fig. 4. Antarctic melt lakes were generally shallower than those seen on Greenland (Table 3) - only lake 4a exceeded 3 m in depth, whereas the mean maximum depth over Greenland was 4.08 m. The heterogeneity and noisiness of Amery Ice Shelf retrievals rendered best-fit calculations difficult and increased uncertainty estimates. However, these melt lakes were 3-8 km in extent, thus facilitating detection in histograms. Greenland lakes exhibited a wider range of sizes, but the algorithm successfully performed retrievals for lakes as small as 200 m in extent.

On average, the noisier data from ICESat-2 produces uncertainties greater than 0.2 m for the Antarctic lakes and 0.3 m for the Greenland lakes, as seen in Table 3, Column 8. The inclusion of lower-confidence photons increases uncertainty despite the restricted bed photon criteria, for the larger photon cloud increases the spread of the entire lake profile. The polynomial errors in ICESat-2 are comparable to ATM over Greenland, with lakes 4f and 4g exhibiting notably high error. Polynomial errors are higher for the Antarctic melt ponds, as it is difficult to reproduce the complex bed topography, an observation shared with lakes

4f and 4g from Greenland. It also must be noted that bed photons are more likely to be found in the ATL03 photon cloud than in ATM waveforms, meaning that the polynomial estimates are less necessary for ICESat-2 than when used for ATM.

## 5 Discussion

### 5.1 Algorithm Performance

The conical scanning of the ATM lidar produced oscillations in 1D elevation profiles that dampened over lake surfaces, so

lakes generally were easier to identify with the airborne retrievals. Flights conducted during the OIB campaign actively avoided cloudy conditions, reducing attenuation sources and further simplifying the lake-finding process over common melt regions. The data volume per granule was lower than ATL03, resulting in less time needed to run the algorithm. However, the number of retrievals possible with ATM is limited, so observations with the lidar best serve as a validation and correction tool for ICESat-2 and other retrieval methods.

The laser power and detector sensitivity of the ATLAS instrument on board ICESat-2 are sufficient to reliably detect lake beds, and a high along-track resolution will correspond to improved estimates of lake bed topography, water depth, and water volume. Despite strong advantages, significant difficulties must be considered before automatic lake detection is feasible. At its operational altitude, the ATLAS laser is subject to scattering and absorption by clouds. Clouds are common over the fringes





of Antarctica and Greenland (Bennartz et al., 2013; Lachlan-Cope, 2010; Van Tricht et al., 2016), and often their optical depth is sufficient to render the surface undetectable. Handling the large data volumes in ATL03 granules also presents a significant challenge. A single granule provides coverage over hundreds of kilometers, so the running time of the algorithm increases relative to ATM granules. Lakes smaller than 1 km are difficult to automatically detect with the algorithm, but LSBS may still
be performed for lakes as small as 200 m if the location of a lake is known through other means (e.g., Landsat-8 imagery or ATM retrievals).

We attribute observed differences in lake topography to the underlying ice surfaces. Supraglacial lakes on Greenland typically form into smooth basins within depressions formed by the underlying bedrock, and their location is independent of ice motion (Echelmeyer et al., 1991). In contrast, meltwater on the Amery Ice Shelf originates from the blue ice zone, propagating
along the ice surface in streams. The location of lakes and ice topography are thus tied to the flowlines of the ice shelf surface. These features are flooded in the Antarctic melt season, producing melt lakes and streams up to 80 km in length (Mellor and McKinnon, 1960; Phillips, 1998; Kingslake et al., 2017).

## 5.2 ATL03 Photon Classification Uncertainties

The classification of ATL03 signal photons provides challenges for automatic depth retrievals. The signal confidence for pho-
tons is provided through the "signal_conf_ph" variable, with higher values (at a maximum of 4) indicating greater confidence that a photon is the signal. However, the confidence variable is given as a 5 x N matrix, where each row represents a surface type (land, ice, inland water, etc.). The confidence level for photons varies between surface types (Neumann et al., 2019b), and photons may be assigned to multiple surface types. Furthermore, the water masks used for surface classification do not factor melt lakes, so lakes on ice-sheet surfaces are not identified. Instead, lake surfaces in ATL03 typically classify as "land ice", and
lake beds as either "land ice" or "land".

The example in Fig. 5 shows the differences between the two surface types. The land-only photons identify the lake surface and portions of the lake bed with high confidence, whereas the lake bed exhibits lower confidence in the land ice classification. For lake beds, photons >2 m below the surface are more likely to be assigned the "land" classification. We theorize this occurs because the lake resembles a tree canopy in 1-D profiles. To circumvent this issue, the LSBS algorithm includes a routine that
takes the highest confidence level among all surface types for each photon in a data granule.

By default, the LSBS algorithm incorporates high-/medium-/low-confidence signal photons and "buffer" photons. This approach is necessary for most melt lakes observed by ICESat-2, as lake beds frequently receive lower confidence flags. For instance, melt lake 4a features an incomplete bed when only high confidence photons are considered. The inclusion of low-/medium- confidence and "buffer" photons reveals a complete bed profile, increasing the maximum depth estimate from 2.60
30  m to 4.57 m, as seen in Fig. 5a. Thus, it is recommended to include photons of multiple confidence levels, otherwise the lake bed may appear attenuated and hinder retrieval efforts.





## 6 Conclusions

We present a method to detect supraglacial lakes and estimate lake depth from 532 nm laser altimetry data. We establish test cases for lake detection over two regions of Greenland (Hiawatha Glacier, 19 July, 2017 and Jakobshavan Isbræ, 17 June, 2019) and East Antarctica (Amery Ice Shelf, 2 January, 2019), and our results demonstrate that depth retrievals are possible using

laser altimetry. Verification of lake detection is given with lake surface flatness tests, where we observe low topographical variance over lake surfaces relative to surrounding ice. Lake bottoms are easy to identify once lake surfaces are established, given that the lakes are not too deep.

We introduce lake a surface-bed separation scheme for ATM and ICESat-2 geolocated photon data to determine the maximum depth of lakes. Our results indicate that altimetry signals reliably detect bottoms as deep as 7 m, after which absorption

of the photons in water reduces the number of reflected photons. Heterogeneity at the lake bed also produces attenuation, complicating retrieval attempts for lakes with rough bed topography or with high impurity concentration. Additional work is required to assess the impacts of lake impurities and geometry on altimetry signals and to improve estimates for such cases. Despite these shortcomings, we anticipate retrieval capability to improve as observations from the 2019 and 2020 Arctic melt seasons are released.

We establish the feasibility for estimates of supraglacial lake depth over Antarctica and Greenland. The high accuracy of 532 nm laser altimeters allow these results to serve as a benchmark for future retrieval studies. Future studies need to examine the accuracy of ICESat-2 lake retrievals relative to ATM where applicable, with additional comparisons to depth estimates from passive imaging sensors.

*Competing interests.*  The authors declare that they have no conflict of interest.




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



**Figure 1.** True-color Landsat-8 composites of Hiawatha Glacier on 18 July, 2017 (a), the Amery Ice Shelf on 1 January, 2019 (b), and the western Greenland ablation zone on 17 June, 2019 (c). Flight tracks for Operation IceBridge (a) and ICESat-2 (b, c) are shown in dotted orange.



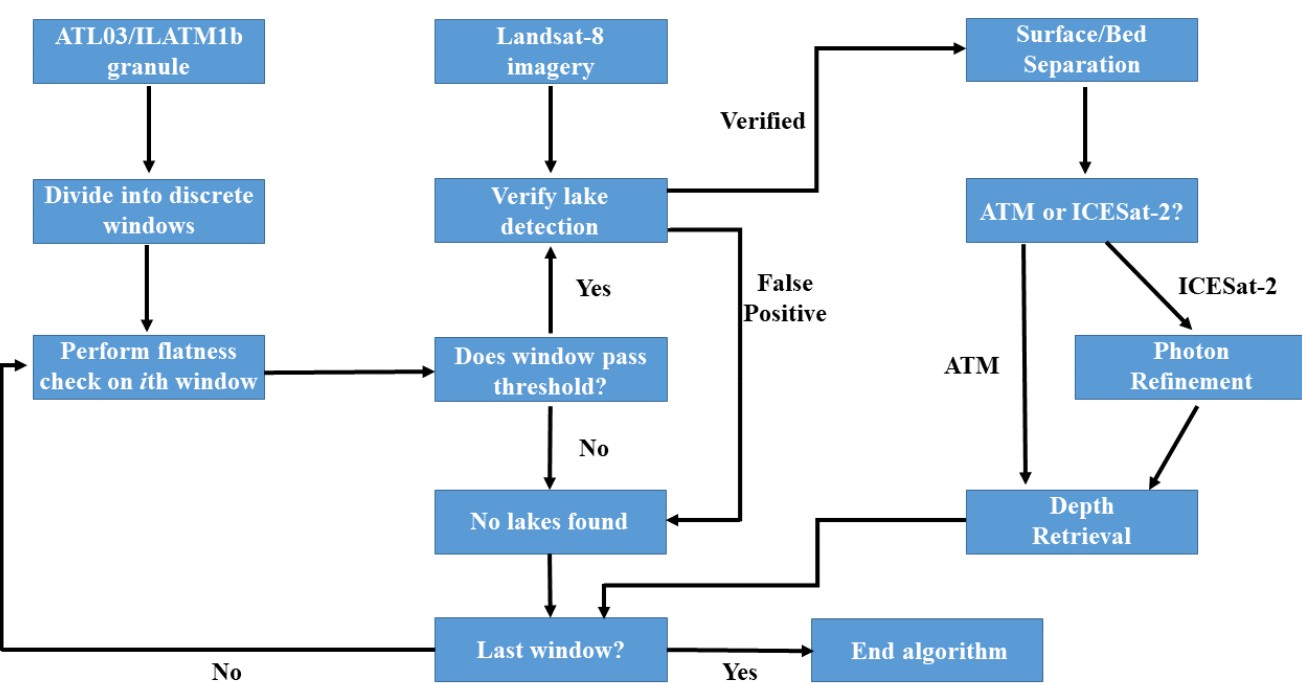

**Figure 2.** Schematic for the workflow of the lake surface-bed separation algorithm.

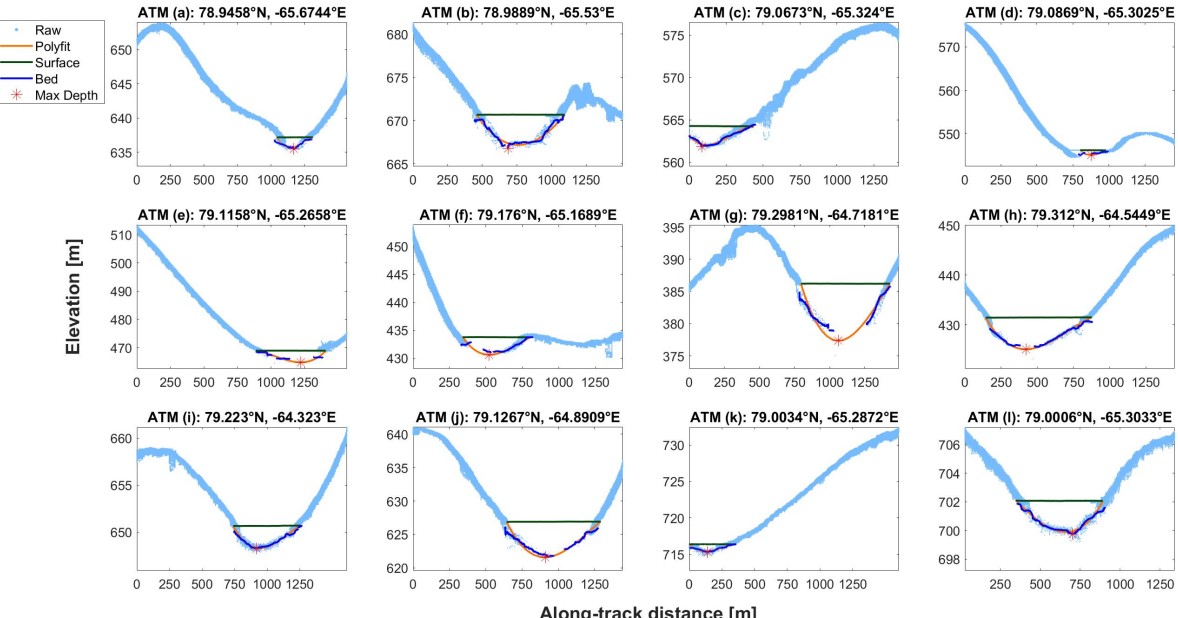

**Figure 3.** ATM lake profiles from 17 July, 2017 fitted using lake surface-bed separation, including the raw ILATM1B product, the lake surface signal, the lake bottom signal, the polynomial-fitted bottom, and the point of maximum depth. Along-track distance is relative to the start of a data granule.

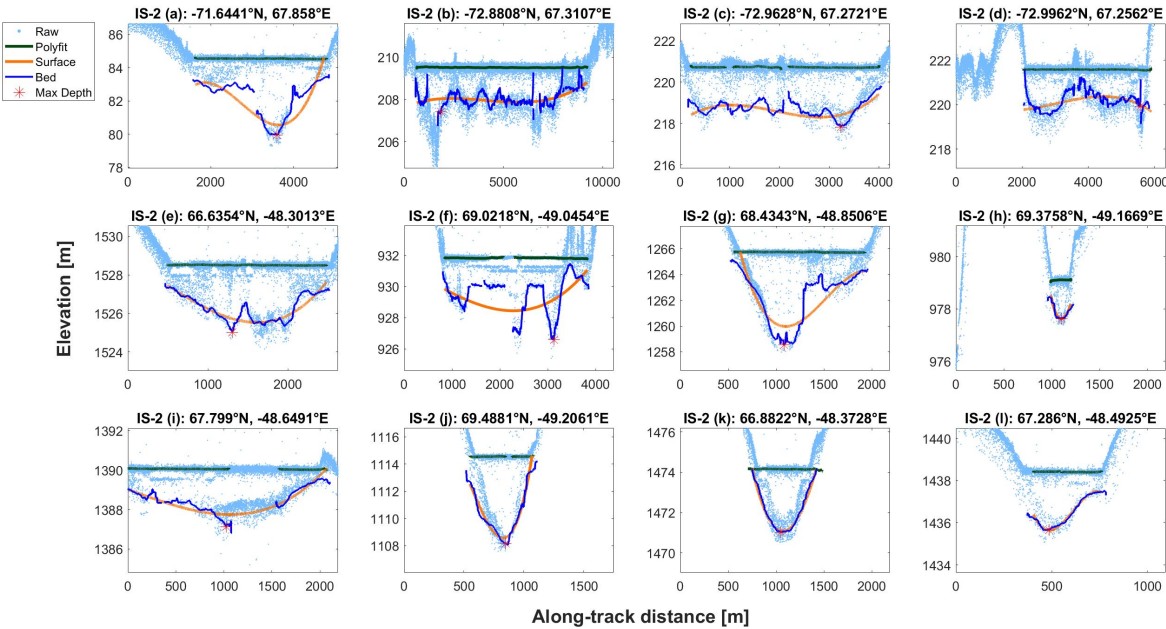

**Figure 4.** Supraglacial lakes and melt ponds detected by ICESat-2 over the Amery Ice Shelf (a-d, first observed by Magruder et al. (2019)) and western Greenland (e-l), using Tracks 0081 and 1222, respectively.





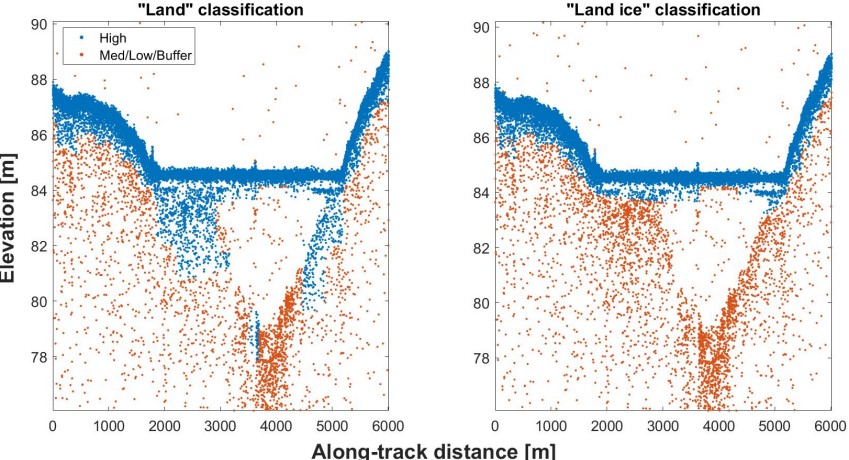

**Figure 5.** Differences in signal confidence between the "land" and "land ice" surface classifications in the ATL03 algorithm. In both examples, photons received by the lake bottom are labeled as "buffer" photons by the ATL03 algorithm.





**Table 1.** Information on the data used for each altimeter.

| Altimeter | Region | Track(s) | Date |
|---|---|---|---|
| ICESat-2 | Amery Ice Shelf | 0081 | 2 January, 2019 |
| | Jakobshavan Isbræ | 1222 | 17 June, 2019 |
| ATM | Hiawatha Glacier | 135106-144357 | 19 July, 2017 |



**Table 2.** Cumulative statistics for ATM supraglacial lakes explored in this study, including mean and maximum depth for $d_s$ and $d_p$, $L$, mean lake depth uncertainty ($\overline{\sigma}_d$), and mean polynomial estimation error ($\overline{\epsilon}_p$). Units are in meters.

| Lake | $\overline{d}_s$ | $max(d_s)$ | $\overline{d}_p$ | $max(d_p)$ | $L$ | $\overline{\sigma}_d$ | $\overline{\epsilon}_p$ |
|------|------|------|------|------|-----|------|------|
| 3a | 0.98 | 1.69 | 0.91 | 1.51 | 270 | 0.08 | 0.31 |
| 3b | 2.25 | 3.75 | 2.32 | 3.49 | 640 | 0.15 | 0.45 |
| 3c | 1.33 | 2.39 | 1.33 | 2.24 | 440 | 0.09 | 0.25 |
| 3d | 0.64 | 0.98 | 0.71 | 1.09 | 180 | 0.10 | 0.38 |
| 3e | 1.81 | 2.98 | 2.37 | 4.11 | 520 | 0.05 | 0.42 |
| 3f | 1.70 | 2.70 | 1.97 | 3.15 | 470 | 0.10 | 0.49 |
| 3g | 4.32 | 7.38 | 5.50 | 8.83 | 630 | 0.20 | 1.03 |
| 3h | 3.64 | 5.91 | 3.90 | 6.37 | 730 | 0.15 | 0.41 |
| 3i | 1.56 | 2.38 | 1.48 | 2.37 | 510 | 0.12 | 0.15 |
| 3j | 3.17 | 5.18 | 3.39 | 5.29 | 650 | 0.11 | 0.65 |
| 3k | 0.60 | 1.06 | 0.55 | 0.97 | 350 | 0.09 | 0.21 |
| 3l | 1.45 | 2.32 | 1.39 | 2.18 | 590 | 0.11 | 0.15 |
| Mean | 1.95 | 3.23 | 2.15 | 3.47 | 500 | 0.11 | 0.41 |





**Table 3.** As with Table 2, but for ICESat-2 tracks.

| Track | Lake | $\bar{d}_s$ | $max(d_s)$ | $\bar{d}_p$ | $max(d_p)$ | L | $\bar{\sigma}_d$ | $\bar{\epsilon}_p$ |
|---|---|---|---|---|---|---|---|---|
| | 4a | 2.32 | 4.57 | 2.62 | 4.00 | 3170 | 0.25 | 0.60 |
| | 4b | 1.48 | 2.67 | 1.48 | 1.70 | 8570 | 0.80 | 1.25 |
| 0081 | 4c | 2.02 | 2.86 | 2.08 | 2.41 | 3790 | 0.28 | 0.46 |
| | 4d | 1.39 | 2.32 | 1.46 | 1.96 | 3860 | 0.77 | 1.21 |
| | **Mean** | 1.80 | 3.11 | 1.91 | 2.52 | 4850 | 0.53 | 0.88 |
| | 4e | 2.24 | 3.43 | 2.28 | 2.98 | 1990 | 0.28 | 0.44 |
| | 4f | 2.31 | 5.22 | 2.66 | 3.44 | 2980 | 0.26 | 1.07 |
| | 4g | 3.52 | 7.15 | 3.76 | 5.78 | 1370 | 0.49 | 0.94 |
| | 4h | 1.22 | 1.47 | 1.24 | 1.50 | 211 | 0.12 | 0.14 |
| 1222 | 4i | 1.52 | 2.88 | 1.55 | 2.37 | 2070 | 0.23 | 0.37 |
| | 4j | 4.13 | 6.56 | 4.13 | 6.01 | 530 | 0.73 | 0.87 |
| | 4k | 1.65 | 3.13 | 2.04 | 3.08 | 780 | 0.22 | 0.25 |
| | 4l | 1.93 | 2.76 | 1.93 | 2.78 | 360 | 0.15 | 0.21 |
| | **Mean** | 2.32 | 4.08 | 2.45 | 3.49 | 1290 | 0.31 | 0.55 |