# Peer review of "Using ICESat-2 and Operation IceBridge altimetry for supraglacial lake depth retrievals"

_The Cryosphere, 2020_

## Referee Comment (RC1) · Anonymous Referee #1 · 28 Jun 2020

**1 Overview**

Fair et al. (2020) use laser altimetry measurements from ICESat-2 and the Airborne Topographic Mapper to map supraglacial lakes on regions of the Greenland and Antarctic ice sheets. This is one of the first of what will likely be many studies incorporating ICESat-2 geolocated photon data to map supraglacial lake depths. The paper provides a method for extracting lake surface and bottom elevations from 532nm laser altimetry data. The work presented by the authors falls within the scope of *The Cryosphere* and could make an interesting contribution to a developing field of measuring supraglacial lakes with remote sensing data. Overall, while this is a promising methods study of supraglacial lakes with laser altimetry measurements, there are a few issues that

should be resolved before its publication.

**2  Broad comments**

- There are places outlined in the line-by-line comments where it could be more quantitative

- Some of these sections are "in the weeds" concerning HDF5 variable names or classification differences. These sections and figures could probably be excluded

- I would expand on the section of difficulties of making a fully automatic lake depth detection algorithm with laser altimetry

- I would mention the impact of detector saturation on highly flat specular surfaces creating a "false" bottom return. This could be noted in the Algorithm Performance section.

**3  Line-by-line comments**

**Page 1, Lines 2–4:**  I would split this sentence to be something like:
"Detection of lake extent, depth, and temporal evolution is important for understanding glacier dynamics. Previous remote sensing observations of lake depth are limited due to inherent uncertainties of depth retrievals with passive remote sensing techniques, and the high absorption of infrared laser energy in water from the original ICESat mission."

**Page 1, Line 8:**  I would change this to reliably or statistically detect lake beds as deep as 7m.

**Page 1, Line 10:** The insufficient classification of photon events when profiling lakes is expected due to how the ATL03 classification algorithms work with a bimodal or multimodal surface, particularly if the lake surface return is not specular.

**Page 1, Line 16:** You are noting here that the contributions to sea level rise from ice sheets will likely overtake steric sea level effects and not that the contributions will overtake glacier and ice caps correct?

**Page 1, Line 17:** I would probably use "aggregation" and not "accumulation".

**Page 1, Line 18:** "When unfrozen, these lakes exhibit a lower albedo than the surrounding ice,"

**Page 1, Line 21:** "which can lead to potentially significant impacts"

**Page 2, Lines 2–4:** "Meltwater penetration into the ice during catastrophic lake drainage events can also lead to hydrofracture, a mechanism through which meltwater facilitates full ice fracture as a result of the stresses induced by the density contrast between liquid water and ice"

**Page 2, Line 5:** "thus can impact sliding velocity and ice discharge"

**Page 2, Line 12–13:** Hopefully we don't reach a time where supraglacial lakes are present over the entirety of either ice sheet. "sheer size of the ice sheet ablation areas"

**Page 3, Line 4:** "6 distinct beams named in the products based on the ground track: GT1L/R, GT2L/R, and GT3L/R".

**Page 3, Line 6:** "approximately every 0.7 meters"

**Page 3, Line 16:** GT2R can be either the central strong beam or the central weak beam based on the orientation of the spacecraft. For the both of your dates (2019-01-02 and 2019-06-17) GT2R was the weak beam.
**Page 3, Lines 18–20:** This is expected due to transmit pulse truncation. The transmit pulse shape is slightly non-Gaussian with a trailing tail. Calculating the average of photon events without that trailing tail biases the results compared to a "true" surface.

**Page 3, Lines 21–22:** Versions of ATM have flown in Greenland since 1993. As written it suggests that ATM was designed as a gap filling instrument rather than an existing and verified instrument suite used in this role.

**Page 3, Line 25:** The ATM1B QFIT elevation product is not a geolocated photon product but a geolocated elevation product

**Page 3, Line 26:** While ATM does not contain a statistical confidence definition, ATM uses a thresholded centroid model from their digitized waveforms and thus will typically only retrieve higher confidence returns. The data is also processed prior to release for QA/QC purposes.

**Page 3, Line 26:** Remove "Despite this"

**Page 3, Lines 27–28:** "Here, the ATM results serve as a proof of concept for the lake detection algorithm"

**Page 4, Lines 4–5:** The lake surfaces aren't necessarily "easily" identifiable and potential lake beds can be hard to detect on highly flat surfaces because detector saturation (related to first-photon-bias) can lead to a non-existent false bottom.

**Page 4, Lines 5–6:** "To simplify the identification of lake features, we separated them into two arrays: one for the surface and one for the bed, which we refer to as "lake surface-bed separation" (LSBS)"

**Page 4, Line 8:** What is $\sim 10^4$–$10^5$ photons in terms of distance?

**Page 4, Line 11:** Are there times when the lake bottom can be the dominant return?

**Page 4, Line 12:** "We check the flatness of the window by computing the standard deviation"

**Page 4, Lines 23–24:** Seems somewhat arbitrary that the thresholding needed to be different surface classification. Would it be better to only use the full set of potential signal photons and the second set of thresholds?

**Page 4, Line 30:** I would say that these were "potential" or "probable" false positives

**Page 5, Line 6:** "overlapping 40 meter segments"

**Page 5, Paragraph 2:** The ATL06 algorithm assumes a single returning surface within a segment of photon events. In supraglacial lake instances, the ATL06 algorithm can compute a height for either lake bottom or lake surface depending on their corresponding return strength. These return strengths can be highly variable.

**Page 5, Line 18:** The ATL06 algorithm uses 3m as the minimum window height.

**Page 5, Line 26:** I would mention that 3cm is far below the horizontal geolocation uncertainty of ICESat-2

**Page 6, Line 10:** What do you mean quantitatively by agree well?

**Page 6, Lines 12–14:** These sentences are awkwardly phrased.

**Page 6, Lines 16–18:** You list the dates here, in Figure 1 and in Table 1. I would get rid of Table 1 as it seems extraneous.

**Page 6, Lines 21–23:** Why mention this?

**Page 6, Lines 25–26:** I would rewrite to be "We detected 12 lakes with sufficient bed returns from the ATM data and 16 potential lake surfaces overall."

**Page 6, Line 27:** What do you mean quantitatively by good accuracy?

**Page 7, Line 3:** "lake bed elevations"

**Page 7, Line 4:** "perhaps influenced by low signal-to-noise ratios or the conical scanning of the lidar instrument"

**Page 7, Lines 8–9:** "We examined an additional 12 supraglacial lakes with ICESat-2, eight in Greenland and four highlighted in Magruder et al. (2019) on the Amery Ice Shelf in Antarctica (Fricker et al., in prep.)."

**Page 7, Line 10:** What do you mean by reasonable success?

**Page 7, Line 21:** less necessary for ICESat-2 than ATM for the supraglacial lakes studied here

**Page 7, Line 34:** ICESat-2 returns are also affected by first-photon-bias (particular if complete saturation of the detectors occurs), blowing snow events (which by forward scattering can create sub surface photons or have a multi-modal return by the snow itself), and solar radiation background.

**Page 8, Line 7:** What do you mean in the attribution sentence?

**Page 8, Lines 14–31:** I'm not sure if mixing classifications is the best approach for determining the signal classification. You're right that supraglacial lakes are not quite fit in any default category in the signal classification algorithm of ATL03. I may be mistaken but I think supraglacial lakes being classified higher in "land" than "land ice" makes sense due to the tighter histogram window of "land ice" (and not that supraglacial lakes resemble canopies). Going forward, it might be better to use signal and buffer photons of a single surface class and iterate to remove potential background photons.

**Page 9, Line 7:** Should add a value for "too deep"

**Figure 4:** The polynomial fits are pretty poor for complex beds. I get the need to not overfit the beds, but would it be better to use a variable order of polynomial or splines?

**Figure 4:** Bed detection seems to be a bit off on the lake edges (a, e, g, i, j, k, l)

**Figure 5:** I don't know if this figure has much meaning.

**Table 1:** I don't think this table is necessary with Figure 1 and the text.

**References**

Z. Fair, M. Flanner, K. M. Brunt, H. A. Fricker, and A. S. Gardner. Using ICESat-2 and Operation IceBridge altimetry for supraglacial lake depth retrievals. *The Cryosphere Discussions*, 2020: 1–21, 2020. doi: `10.5194/tc-2020-136`.

H. A. Fricker, P. Arndt, S. Adusumilli, K. M. Brunt, T. Datta, Z. Fair, M. Jasinski, J. Kingslake, L. Magruder, M. Moussavi, A. Pope, and J. J. Spergel. Revisiting surface meltstreams on Amery Ice Shelf, East Antarctica, in prep.

L. Magruder, T. Neumann, H. Fricker, S. Farrell, K. Brunt, A. Gardner, D. Hancock, K. Harbeck, M. Jasinski, R. Kwok, N. Kurtz, J. Lee, T. Markus, J. Morison, A. Neuenschwander, S. Palm, S. Popescu, B. Smith, and Y. Yang. New Earth Orbiter Provides a Sharper Look at a Changing Planet. *Eos*, 100:n/a–n/a, Sept. 2019. ISSN 2324-9250. doi: `10.1029/2019EO133233`.

---

## Referee Comment (RC2) · Allen Pope (Referee) · 6 Jul 2020

This paper provides straightforward information about a very interesting proof-of-concept application of ICESat-2 to studying supraglacial lake depth, which I am sure will be the basis of much future work and investigation. Thank you for this work, which I'm sure will push the work of our research community forwards! It is well written, generally easy to follow, and provides robust conclusions it its scope. In this review I have only a few comments and questions that I hope will bring even more clarity and ability for further applications - some on style & methods, some on date & code citation/sharing, and some about the particular scope that this paper has chosen.

Scope: *The paper limits itself to central strong beam (GT2R), but then also includes

lower confidence photos from this band which "decreases measurement precision but gives better agreement with ground-based data." Because of this, I am wondering why other beams were not used, or at least their potential use discussed in the paper? *Page 6 Line 14: Do you have any estimate for just how widely applicable these methods will be / how easy it is to get good coverage? I understand you have to put limits on this paper somewhere, for sure, so this is mostly out of curiosity and might be of interest in a discussion/conclusion?

Data & Code Citation/Sharing: **The Cryosphere's data policy states that "Authors are required to provide a statement on how their underlying research data can be accessed. This must be placed as the section "Data availability" at the end of the manuscript." I did not see such a section. Clarity in citing the exact subsets of the large datasets that you cite would be ideal (which I know is also in your Table 1, but not presented in one place).

**The Cryosphere guidelines also state that "Data do not comprise the only information which is important in the context of reproducibility. Therefore, Copernicus Publications encourages authors to also deposit software, algorithms, model code, video supplements, video abstracts, International Geo Sample Numbers, and other underlying material on suitable FAIR-aligned repositories/archives whenever possible. These materials should be referenced in the article and cited via a persistent identifier such as a DOI." There is clearly a lot of important code developed and used by the authors, and it would be in line with this journal's goals that it be documented, shared, and cited. This would allow for reproducibility, further application of these methods, and further refinement, as well. I very much hope that the reviewers document, share, and cite the final version of their code to make their methods as open as the data they use and the publication they have chosen to publish in. (Of course, if you have another code/methods paper in prep, then please do cite that and I apologize for jumping the gun!)

*And since I'm talking about data and code sharing - at the risk of inviting my own

citation - you cited Pope at al 2016 on Page 2 Line 23/24. I wonder whether you might (also) want to cite Pope (2016), which I bring up here because if more fully describes, documents, and shares the code developed and used in the Pope et al paper. https://doi.org/10.1002/2015EA000125

Other Manuscript Comments: *Page 1 Line 8-9 (Abstract): Can you quickly mention where the uncertainties are derived from here? It might just be me, but if quickly reading, it makes it sounds like there is comparison to some in situ data...

*Page 4 Line 10: about how long is each data granule, in ground distance, to include $10^4$ - $10^5$ photos per window? I think this will help people understand the next assumptions.

*Page 4 line 14: How were these ranges selected / chosen? This would seems to be an important part of method development.

*Page 4 Line 14: I'm sorry if I missed it, but can you define sigma in the text upon first usage?

*Page 4 Line 17: Consider replacing "lake surface" with "height of the lake surface" and underlining the letters h, s, f, and, c in order to make the abbreviation very clear?

*Page 4 Line 20: How were these ranges selected / chosen? This would seems to be an important part of method development.

*Page 4 Line 22: Consider replacing "lake surface" with "height of the lake bottom" and underlining the letters h, b, t, and, m in order to make the abbreviation very clear?

*Page 4 Line 27: How were these filters chosen? This would seems to be an important part of method development.

*Page 4 / Section 3.1 in general: It would be even clearer to present these methods if there were agreement between the steps here and in Figure 2 (e.g. one box / arrow per bullet point).

*Page 6 Line 18: I could be wrong, but it is possible that Figure 1 and 2 are cited in backwards order? You might consider flipping their numbers?

*Page 5 Line 27: You mention a refraction correction but then there is no further detail. I know it is pretty basic, but for full clarity perhaps describe slightly more / provide a citation for the method you use for refraction correction?

*Page 6 Line 29: I wonder if you this it is important to re-emphasize the filtering of which lake depths were kept in presenting average lake depths? e.g. lots of shallow lakes aren't being included?

*Figure 1: Consider using dots to indicate location, rather than ovals, which are much larger than the image are?

*Figure 2: It is slightly confusing that you use the same blue boxes for both data and processes (e.g. Landsat 8 imagery vs verify lake detection), consider using different shapes / colors / some design choice to indicate the difference?

*Table 2: Please also define $d_s$, $d_p$, and L in the table caption.

[Figure]

---

## Author Comment (AC1) · 19 Aug 2020

Response to Reviewer #1 on the manuscript:

*Using ICESat-2 and Operation IceBridge altimetry for supraglacial lake depth retrievals* by Fair et al.

We thank the reviewer for their comments and suggestions to improve the clarity and structure of the manuscript. In this response, the original comment is given in black, the authors' response in blue, and the proposed changes in orange.

Broad comment (1) "There are places outlined in the line-by-line comments where it could be more quantitative"

We addressed these suggestions on a case-by-case basis, as seen in the line-by-line responses below.

Broad comment (2) "Some of these sections are 'in the weeds' concerning HDF5 variable names or classification differences. These sections and figures could probably be excluded."

We agree that Section 5.2 is too esoteric and technical in its current form. We assert, however, that this paper is a proof-of-concept study designed to highlight potential issues that a user may experience when performing automated supraglacial lake depth retrievals from ICESat-2 data. Therefore, while we propose to rewrite Section 5.2 to be less technical, we will retain a brief mention of the photon classification issues. The proposed rewrites are expanded upon in the responses to broad comments (3) and (4).

Broad comment (3) "I would expand on the section of difficulties of making a fully automatic lake depth detection algorithm with laser altimetry."

As noted above, we plan to rewrite Section 5.2 to focus on the difficulties of automated retrievals from ICESat-2 data (Fricker et al., in prep.), with examples from the data formatting (i.e. photon classification) and from lake properties (bed topography, size, etc.), as follows:

**Section 5.2    Automation Challenges**

The identification of lake beds in the LSBS algorithm is based on a window of acceptable photons. The photon window is constrained by the coefficients $a$ and $b$ (for ICESat-2, $a = 1.0$, $b = 0.5$). Lake beds detected in this manner had a height uncertainty of 0.38 m (Table 2). The coefficients for ATM ($a = 1.8$, $b = 0.75$), resulted in more accurate retrievals on an individual basis. However, implementing varying $a$ and $b$ values proved difficult to automate, as other values may produce more accurate depths.

The challenges in full automation are related to three key issues. First, the observed extent of lakes varied considerably, especially over Greenland. The diversity in lake sizes complicated attempts to derive a universal "flatness" check. Smaller lakes present fewer lake surface photons, so a smaller data window (~$10^4$ photons) is required to prevent false positives. However, larger lakes may not be fully represented in smaller windows. A larger data window (~$10^5$ photons) will fully capture the largest lakes, but smaller lakes may then be overlooked.

Second, multiple scattering at the lake bed increases the photon spread and thus also increases the uncertainty of depth retrievals. Most supraglacial lakes observed by ATM featured smooth beds, so photons experienced one or few scattering events before returning to the detector. The instrument digitizer automatically filters return signals with low photon counts, reducing the spread of bed photons, at the cost of deep lake bottom detection. In contrast, the lakes observed with ICESat-2 exhibited more heterogeneous beds, leading to increased scattering events by photons and delays in return pulses. In these cases, the given values for $a$ and $b$ may not produce the most accurate bed solution. Furthermore, if the return is significant for a given photon window, then it may lead to a false negative for a portion of the lake (Figure 4i). To reduce uncertainty in lake depth retrievals, future improvements in working with ICESat-2 data should focus on identifying and filtering multiple scattering.

Finally, the ATL03 signal-finding algorithm is considered conservative in that it accepts false positives (background photons classified as signal photons) to ensure that all signal photons are passed to higher-level products (Neumann et al., 2020). Thus, uncertainties in the ATL03 photon classification contribute to noise in the water column and the lake bed. The classification algorithm uses surface masks to allocate statistical confidence to ATL03 photons for multiple surface types (Neumann et al., 2019b), with overlap possible among the surfaces. Melt lakes are too short-lived to be considered "inland water" and are instead categorized as "land ice" (lake surface) and "land" (lake surface and bed). Because the "land" classification also includes the bed, it includes more potential signal photons than land ice, so we recommend to only use land photons when performing supraglacial lake depth retrievals. It must be noted, however, that a lake bed profile is fully resolved only with the inclusion of low-/medium- confidence and "buffer" photons. The buffer photons ensure that all photons identified as surface signal are provided to the appropriate upper-level data product algorithms. However, they can introduce more noise to the profile, so more sophisticated filtering techniques are needed to distinguish the signal photons against the solar background.

Broad comment (4) "I would mention the impact of detector saturation on highly flat specular surfaces creating a "false" bottom return. This could be noted in the Algorithm Performance section."

We agree that it is important to note the potential effects of specular reflection on observed lake surfaces. We will note its impact on lake depth retrievals in Section 5.1 as a potential obstacle.

A potential issue for lake depth retrievals concerns specular reflection. When photons interact with a flat water surface, they may reflect directly back to the detector with minimal energy loss.

The excessive return energy produces a "dead time" in the ATLAS detector, and the return signal is represented by multiple subsurface returns approximately 2.3 m and 4.2 m below the true surface (Neumann et al., 2020). An example of this phenomenon may be seen in Fig. 4f, where a prominent subsurface return is featured along the lake extent. However, because the subsurface echo is smaller than the true surface when viewed through histograms, the LSBS algorithm is able to avoid biases caused by specular reflection.

Page 1, Lines 2-4: "I would split this sentence to be something like: 'Detection of lake extent, depth, and temporal evolution is important for understanding glacier dynamics. Previous remote sensing observations of lake depth are limited due to inherent uncertainties of depth retrievals with passive remote sensing techniques, and the high absorption of infrared laser energy in water from the original ICESat mission.'"

We changed as requested with a slight edit, as we do not see it necessary to mention ICESat here:

…for understanding glacier dynamics. Previous remote sensing observations of lake depth are limited to estimates from passive satellite imagery, which has inherent uncertainties, and there is little ground truth available.

Page 1, Line 8: "I would change this to reliably or statistically detect lake beds as deep as 7m."

We made the following change:

Both lidars reliably detect lake beds as deep as 7 m.

Page 1, Line 10: "The insufficient classification of photon events when profiling lakes is expected due to how the ATL03 classification algorithms work with a bimodal or multimodal surface, particularly if the lake surface return is not specular."

The statement given here is addressed to a general scientific audience that may not be aware of the ATL03 classification algorithms. However, we agree it is to be expected, so we will reword the text as follows:

The bimodal nature of lake returns means that high-confidence photons are often insufficient to fully profile lakes, so lower confidence and buffer photons are required to view the lake bed.

Page 1, Line 16: "You are noting here that the contributions to sea level rise from ice sheets will likely overtake steric sea level effects and not that the contributions will overtake glacier and ice caps correct?"

You are right. To make this clearer, we reworded the statement as:

…leading to the contributions from both ice sheets to overtake the contribution of thermal expansion to sea level rise (Vaughan et al., 2013).

Page 1, Line 17: "I would probably use 'aggregation' and not 'accumulation'".

We changed as requested:

Meltwater plays vital roles in ice sheet evolution […], including aggregation on ice sheets as supraglacial lakes.

Page 1, Line 18: "When unfrozen, these lakes exhibit a lower albedo than the surrounding ice,"

We changed as requested:

When unfrozen, these lakes exhibit a lower albedo than that of the surrounding ice…

Page 1, Line 21: "which can lead to potentially significant impacts"

We changed as requested:

...their spectral emissivity in the IR spectrum also differs from bare ice […], which can lead to potentially significant impacts…

Page 2, Lines 2–4: "Meltwater penetration into the ice during catastrophic lake drainage events can also lead to hydrofracture, a mechanism through which meltwater facilitates full ice fracture as a result of the stresses induced by the density contrast between liquid water and ice"

We applied the following change:

During catastrophic lake drainage events, meltwater penetration into the ice can also lead to hydrofracture…

Page 2, Line 5: "thus can impact sliding velocity and ice discharge"

We changed as requested:

…which in turn modify the resistance to ice flow and thus can impact sliding velocity and ice discharge.

Page 2, Line 12–13: "Hopefully we don't reach a time where supraglacial lakes are present over the entirety of either ice sheet. 'sheer size of the ice sheet ablation areas'"

We agree. We applied the following change:

However, the harsh conditions of Antarctica and Greenland, the transience of meltwater, and the sheer size of the ice sheet ablation zones…

Page 3, Line 4: "6 distinct beams named in the products based on the ground track: GT1L/R, GT2L/R, and GT3L/R."

We applied the following change:

…a 532 nm micro-pulse laser that is split into six distinct beams with names based on the ground track…

Page 3, Line 6: "approximately every 0.7 meters"

We changed as requested:

…ICESat-2 records a unique laser pulse approximately every 0.7 m…

Page 3, Line 16: "GT2R can be either the central strong beam or the central weak beam based on the orientation of the spacecraft. For the both of your dates (2019-01-02 and 2019-06-17) GT2R was the weak beam."

You are correct. We edited the text to fix this mistake:

Of the six beams available, we concentrated on the central strong beam (GT2L)…

Page 3, Lines 18–20: "This is expected due to transmit pulse truncation. The transmit pulse shape is slightly non-Gaussian with a trailing tail. Calculating the average of photon events without that trailing tail biases the results compared to a 'true' surface."

We respond to the reviewer assuming that they are referring to the inclusion of lower confidence photons. With this assumption, we added the following:

The addition of medium, low, and "buffer" photons slightly decreases measurement precision, but a less truncated transmit pulse gives better agreement with ATL06 and ground-based data (Brunt et al., 2019b).

Page 3, Lines 21–22: "Versions of ATM have flown in Greenland since 1993. As written it suggests that ATM was designed as a gap filling instrument rather than an existing and verified instrument suite used in this role."

We agree. To make it clearer that we are referring to the IceBridge measurements, we applied the following changes:

The Operation IceBridge (OIB) campaign was designed to fill the gap in polar altimetry between ICESat and ICESat-2. Its scientific payload included the Airborne Topographic Mapper, a 532 nm lidar that has been used for ice sheet and shallow water measurements since 1993.

Page 3, Line 25: "The ATM1B QFIT elevation product is not a geolocated photon product but a geolocated elevation product"

You are correct. We applied the following change:

The ATM Level-1B Elevation and Return Strength (ILATM1B) product converts analog waveforms into a geolocated elevation product…

Page 3, Line 26: "While ATM does not contain a statistical confidence definition, ATM uses a thresholded centroid model from their digitized waveforms and thus will typically only retrieve higher confidence returns. The data is also processed prior to release for QA/QC purposes."

The thresholding applied to ATM data is briefly mentioned in Section 4, but it is in the context of poor signal return for deep lake beds. To highlight the benefits of the centroid model, we added information to Section 5.2 (see response to broad comment #3), and reworded Line 26 as:

...to emulate ATLAS data (Studinger, 2013, updated 2018). Although it lacks a statistical confidence definition, ATM applies a centroid model to digitized waveforms and to retrieve high-confidence photons.

Page 3, Line 26: "Remove 'Despite this'"

We changed as requested:

…retrieves high-confidence photons. Brunt et al. (2019a) found that…

Page 3, Lines 27–28: "Here, the ATM results serve as a proof of concept for the lake detection algorithm"

We changed as requested:

Here, the ATM results serve as a proof-of-concept for the lake detection algorithm.

Page 4, Lines 4–5: "The lake surfaces aren't necessarily "easily" identifiable and potential lake beds can be hard to detect on highly flat surfaces because detector saturation (related to first-photon-bias) can lead to a non-existent false bottom."

We acknowledge that lake surfaces can be difficult to discern from other features (i.e. smooth ice in close proximity to the lake, so we removed the assertion that surfaces are "easily" identifiable. We elaborate on the effects of specular reflection in the revised Section 5 (see above).

…with the expectations that (i) lake surfaces would be identifiable in photon histograms and (ii) lakes may be found via statistical inference in the region of the lake surface.

Page 4, Lines 5–6: "To simplify the identification of lake features, we separated them into two arrays: one for the surface and one for the bed, which we refer to as 'lake surface-bed separation (LSBS).'"

We changed as requested:

…one for the surface and one for the bed, which we refer to as "lake surface-bed separation" (LSBS).

Page 4, Line 8: "What is $\sim 10^4$–$10^5$ photons in terms of distance?"

The given number of photons is equivalent to ~1-10 km in along-track distance for ICESat-2 and ~0.15-1.5 km for ATM. We added the following to provide context:

We divided each data granule into discrete along-track windows to reduce the data volume to $\sim 10^4$-$10^5$ photons per window. This photon count is equivalent to ~1-10 km in along-track distance for ICESat-2 and ~0.15-1.5 km for ATM.

Page 4, Line 11: "Are there times when the lake bottom can be the dominant return?"

Yes - this issue may be observed in Figure 4i, where a strong bottom return in a shallow lake leads to a false negative over part of the lake. We added commentary on this observation in Section 5.

The given values for $a$ and $b$ may not produce the most accurate lake bed solution in these cases. Furthermore, if the bottom return is significant for a given photon window, then it may lead to a false negative for that portion of the lake (Figure 4i).

Page 4, Line 12: "We check the flatness of the window by computing the standard deviation"

We propose this change, as requested:

We check the flatness of the window by computing the standard deviation of high-confidence signal photons…

Page 4, Lines 23–24: "Seems somewhat arbitrary that the thresholding needed to be different surface classification. Would it be better to only use the full set of potential signal photons and the second set of thresholds?"

We agree that ICESat-2 photons do not require thresholds based on classification. Most lakes we considered required medium-/low- confidence and buffer photons, so we applied the second threshold to all cases. The statistics given in Table 3 reflect this thresholding.

However, the second threshold proves ineffective for shallow lakes in ATM data, whereas the first threshold was applicable in all cases. Therefore, to improve readability, we propose to remove the procedural step starting at Line 23, and revised Lines 19-22 to be:

Within these horizontal bounds, photons were defined as a lake bottom if they satisfied the condition: $h_{sfc} - a\sigma_{sfc} \leq h \leq h_{sfc} - b\sigma_{sfc}$, where $\sigma_{sfc}$ is the standard deviation of lake surface photons. The constraints $a$ and $b$ were derived through trial-and-error, such that $a = 1.0$ (1.8) and $b = 0.5$ (0.75) for ICESat-2 (ATM). We set these constraints to reduce the impacts of multiple scattering and specular reflection on depth estimates.

Page 4, Line 30: "I would say that these were 'potential' or 'probable' false positives."

We propose to apply the following change:

If the number of bed photons was very small (100 or less), then the scene was marked as a probable false positive.

Page 5, Line 6: "overlapping 40 meter segments"

We changed as requested:

…ATL03 photon aggregates within overlapping 40 m segments…

Page 5, Paragraph 2: "The ATL06 algorithm assumes a single returning surface within a segment of photon events. In supraglacial lake instances, the ATL06 algorithm can compute a height for either lake bottom or lake surface depending on their corresponding return strength. These return strengths can be highly variable."

We thank the reviewer for providing their knowledge of the ATL06 algorithm. In response, we revised Paragraphs 2 and 3 in the following manner:

The linear regression in ATL06 accounts for all ATL03 photons (background or signal), and the technique performs a background-corrected spread estimate to narrow the range for acceptable photons. This is an iterative scheme; the refinement process repeats its acceptable photon filter

until no photons are removed. As a consequence, the ATL06 algorithm assumes a single returning surface, so over a melt lake it will compute a height for either the lake bottom or the lake surface, depending on their return strengths.

The condition for acceptable surface photons in ATL06 is given by:

$$|r - r_{med}| < 0.5 H_w$$

Within a 40 m photon segment, $r$ is the residual of a photon relative to the linear regression, $r_{med}$ is the median residual, and $H_w$ is window height. The height of the window is taken as the maximum of the observed photon spread, the window height (if any) and 3 m, and photons within the window range are defined as the surface. The LSBS algorithm follows a similar procedure, but the flatness of the lake surface and relatively low photon density of the corresponding beds rendered iterating unnecessary. The lake bed is then defined as photons not within the window and below the surface…

Page 5, Line 18: "The ATL06 algorithm uses 3m as the minimum window height."

You are correct. We applied the following correction:

The height of the window is taken as the maximum of the observed photon spread, the previous window height (if any), and 3 m…

Page 5, Line 26: "I would mention that 3cm is far below the horizontal geolocation uncertainty of ICESat-2"

To highlight the accuracy of ICESat-2 geolocation, we applied the following change:

The central strong beam for ICESat-2 is near-nadir, so the horizontal offset was determined to be small relative to the size of lakes (~3 cm, far below the horizontal geolocation uncertainty of ICESat-2).

Page 6, Line 10: "What do you mean quantitatively by agree well?"

We acknowledge that a quantitative representation of the agreement would be beneficial. We therefore made the following change:

We compare $z_s$ and $z_p$ over lakes with well-defined bottoms, and show in Sect. 4 that the two generally agree to within 0.88 m.

Page 6, Lines 12–14: "These sentences are awkwardly phrased."

We reworded the given sentences in the following manner:

We acknowledge the desire to retrieve lake volume from laser altimetry, but we leave the development of such an algorithm for a future study. For example, depth retrievals from ICESat-2 could potentially be combined with lake radius and shape estimations determined from visible satellite imagery to derive water volume.

Page 6, Lines 16–18: "You list the dates here, in Figure 1 and in Table 1. I would get rid of Table 1 as it seems extraneous."

We assessed the necessity of Table 1, and we agree that it is redundant with the information given in Section 3.4. The table is deleted in the revised manuscript. Lines 19-20 were changed to reflect this:

Comparisons between Landsat-8 imagery and ICESat-2/OIB flight tracks confirmed supraglacial lake overpasses for study. In Spring 2019…

Page 6, Lines 21–23: "Why mention this?"

At the time of writing, ATM data from the 2019 Spring campaign were unobtainable despite the availability of ICESat-2 data. Lines 21-23 are given as justification for the lack of case studies co-located by ICESat-2 and ATM. With the release of the 2019 Spring data, we were unable to find suitable lake candidates that may be observed by both ATM and ICESat-2. We updated the given lines to reflect this:

There were no lakes sampled at the time by both ICESat-2 and OIB.

Page 6, Lines 25–26: "I would rewrite to be 'We detected 12 lakes with sufficient bed returns from the ATM data and 16 potential lake surfaces overall.'"

We changed as requested:

We detected 12 melt lakes with sufficient bed returns from the ATM data and 16 potential melt lake surfaces overall. The melt lake profiles are shown in Fig. 3…

Page 6, Line 27: "What do you mean quantitatively by good accuracy?"

We qualitatively considered the surface accuracy to be "good" if the algorithm distinguished between lake surfaces and the surrounding ice terrain. To provide a more quantitative assessment, we calculated the standard deviation of surface photons ($\sigma_{sfc}$). The mean for $\sigma_{sfc}$ was found to be 0.0087 m, thus demonstrating the effectiveness of the flatness check. We revised Line 27 to reflect this assessment:

The algorithm reliably distinguishes between lake surfaces and the surrounding ice terrain. The mean spread among lake surface photons is 0.0087 m, or well within the flatness threshold of 0.02 m. Lake bottoms are well-defined when $d_s < 8$ m. Lake bottoms deeper than 8 m exhibit fewer signal returns…

Page 7, Line 3: "lake bed elevations"

We changed as requested:

The highest uncertainties are observed for lake depths greater than 3 m…

Page 7, Line 4: "perhaps influenced by low signal-to-noise ratios or the conical scanning of the lidar instrument"

We changed as requested:

...perhaps influenced by low signal-to-noise ratios or the conical scanning of the OIB lidar instrument.

Page 7, Lines 8–9: "We examined an additional 12 supraglacial lakes with ICESat-2, eight in Greenland and four highlighted in Magruder et al. (2019) on the Amery Ice Shelf in Antarctica (Fricker et al., in prep.)."

We applied the following change:

We examined an additional 12 supraglacial lakes with ICESat-2, eight in Greenland and four highlighted in Marguder et al. (2019) and Fricker et al. (in prep) on the Amery Ice Shelf in Antarctica.

Page 7, Line 10: "What do you mean by reasonable success?"

As with Page 6, Line 27, "reasonable success" is a qualitative assessment of the lake surface/bed profiles. We added a quantitative assessment of lake surface/bed uncertainty, as seen below.

The refined algorithm captures lake surfaces and beds reasonably well (Fig. 4), with a mean uncertainty of 0.015 m for surface photons and 0.38 m for bed photons.

Page 7, Line 21: "less necessary for ICESat-2 than ATM for the supraglacial lakes studied here"

We changed as requested:

It must also be noted that lake bed photons are more likely to be found in the ATL03 photon cloud than in ATM waveforms, meaning that the polynomial estimates are less necessary for ICESat-2 than ATM for the supraglacial lakes studied here."

Page 7, Line 34: "ICESat-2 returns are also affected by first-photon-bias (particular if complete saturation of the detectors occurs), blowing snow events (which by for-ward scattering can create sub surface photons or have a multi-modal return by the snow itself), and solar radiation background."

The reviewer is correct. To highlight these issues, we revised Line 33 to include the following:

At its operational altitude, the ATLAS laser is subject to first-photon-bias, solar background radiation, and scattering and absorption by clouds and blowing snow.

Page 8, Line 7: "What do you mean in the attribution sentence?"

In lakes observed by ICESat-2, we observed clear differences in lake bed topography between Antarctica and Greenland. We speculate in the revised Section 5.2 that bed topography may affect the return signal of the ATLAS laser and produce greater uncertainties.

We also realized that the sentence did not flow with the preceding paragraph, so we propose to revise the sentence as follows:

We observed differences in lake topography for ICESat-2 lakes, and we attribute them to the underlying ice surfaces.

Page 8, Lines 14–31: "I'm not sure if mixing classifications is the best approach for deter-mining the signal classification. You're right that supraglacial lakes are not quite fit in any default category in the signal classification algorithm of ATL03. I may be mistaked but I think supraglacial lakes being classified higher in "land" than "land ice" makes sense due to the tighter histogram window of "land ice" (and not that supraglacial lakes resemble canopies). Going forward, it might be better to use signal and buffer photons of a single surface class and iterate to remove potential background photons."

We agree that a single surface class offers a simpler approach to the classification issue discussed in Section 5.2. A follow-up analysis confirmed that the "land" classification is sufficient to characterize the lakes examined in this study. The example given in Figure 5 highlights this point.

According to the ATL03 Algorithm Theoretical Basis Document, classifications are based on surface masks, with overlaps frequent for land and land ice over ice sheets. Therefore, differences between the two surface types should originate from dissimilar criteria for signal

photons, as the reviewer suggests. To amend this misconception, we edited the following for the revised Section 5.2 (see broad comment #3):

Finally, the ATL03 signal-finding algorithm is conservative in that it accepts false positives (background photons classified as signal photons) to ensure that all signal photons are passed to higher-level products. Thus, uncertainties in the ATL03 photon classification contribute to noise in the water column and the lake bed. The classification algorithm uses pre-defined surface masks to allocate statistical confidence to ATL03 photons for multiple surface types (e.g. "*inland water*", "*land ice*", "*land*"; Neumann et al., 2019b), with overlap possible between masks. Melt lakes are categorized as "*land ice*" (lake surface) and "*land*" (lake surface and bed). Because the "*land*" classification also includes the bed, it includes more potential signal photons than *land ice*, so our recommendation is to only use *land* photons for supraglacial lake depth retrievals. It must be noted, however, that a lake bed profile is fully resolved only with the inclusion of low-/medium- confidence and "buffer" photons. The buffer photons ensure that all photons identified as surface signal are provided to the appropriate upper-level data product algorithms. However, they can introduce greater noise to the profile, so more sophisticated filtering techniques are needed to distinguish between signal photons and the solar background.

Page 9, Line 7: "Should add a value for 'too deep'"

We agree. For better consistency with the results, we changed Line 7 to the following:

Lake bottoms are easy to identify once lake surfaces are established, given that the lakes are not deeper than 7 m.

Figure 4: "The polynomial fits are pretty poor for complex beds. I get the need to not overfit the beds, but would it be better to use a variable order of polynomial or splines?"

It is true that the polynomials perform poorly for many of the lakes observed by ICESat-2. The 3rd-order approximation was designed to fill the gaps in deep lakes with the classic "bowl" shape. The polynomial fits therefore perform most effectively for the deep lakes observed by ATM, where the bed topography is less complex.

The lake beds in Figure 4 generally show greater complexity, which results in the poorer fits. In these cases, we agree that a spline interpolation would perform more effectively. We therefore replaced the polynomial fits in Figure 4 with splines on a case-by-case basis. Ultimately, only the polynomial fit in lake 4i was retained due to poor spline fitting.

We highlight the notion that the polynomial/spline fit is less unnecessary for ICESat-2 lakes, given that bed photons are more likely to be found in the ATL03 photon cloud. Its primary function in ICESat-2 retrievals is to fill gaps missed by the initial bed-finding routine, rather than predict the deepest part of lakes. Due to the limited usage of the 3rd-order fit, we removed the "polynomial error" column in Table 3, instead focusing on how the interpolants improved retrievals for lakes 4a, 4b, 4f, and 4i. The updated Figure 4 is shown below.

Lastly, the change to spline fitting for ICESat-2 lakes necessitated alterations to the text. The changes are given here:

Page 6, Paragraph 2: For deep or inhomogeneous lakes, attenuation of photon energy in water resulted in fewer signal photons observed at lake bottoms (Fig. 4). In these situations, we fitted polynomial or spline fits to all lake profiles with bounds at the lake edges. Lakes observed by ATM typically featured "bowl" shapes and attenuation at the deepest parts, so 3rd-order polynomials were sufficient. In ICESat-2 data, the retrieved lake beds showed greater complexity, so we tested polynomial fits and splines on a case-by-case basis. Lake depths approximated with curve fitting were denoted as $z_p$…

Page 7, Lines 17-21: The curve fits improved depth estimates for lakes 4b, 4f, and 4i. Of these lakes, only 4i used a polynomial estimate due to poor spline fitting. The inclusion of interpolants increased the mean depth estimates of 4b, 4f, and 4i by 0.08 m, 0.04 m, and 0.03 m respectively. The spline fitting also significantly increased the maximum observed depth in lake 4b from 2.67 m to 3.27 m. The remaining lakes featured more complete bed profiles, meaning that the fitting estimates were less necessary.

Figure 4: Polynomial curves were replaced with splines. The legend entry "Polyfit" was changed to "CurveFit" (this change also occurred for Figure 3).

Table 3: The polynomial error ($\epsilon_p$) was removed.

[Figure]

Figure 4. Supraglacial lakes and melt ponds detected by ICESat-2 over the Amery Ice Shelf (a-d, reported by Magruder et al., 2019 and Fricker et al., in prep.) and western Greenland (e-l), using Tracks 0081 and 1222, respectively.

Figure 4: "Bed detection seems to be a bit off on the lake edges (a, e, g, i, j, k, l)."

The lake bed detection is restricted to the edges of the lake surfaces. This limitation affects the number of photons considered to be acceptable bed photons, which occasionally leads to a slight skewing of the lake bed near the edges. We observed this issue occurring most frequently with smaller, shallower lakes with fewer photons. The following was added to Lines 9-11 to account for this:

The refined algorithm captures lake surfaces and beds reasonably well (Fig. 4), with a mean uncertainty of 0.015 m for surface photons and 0.38 m for bed photons. The lake edges partially account for the bed photon uncertainty, for the limited number of acceptable photons produces a slight bias in bed estimates.

Figure 5: "I don't know if this figure has much meaning."

After careful consideration, we agreed that Figure 5 is unnecessary for the relevant discussion. We propose to delete it.

Table 1: "I don't think this table is necessary with Figure 1 and the text."

As mentioned in a previous comment, we removed Table 1 and made the appropriate text changes.

---

## Author Comment (AC2) · 19 Aug 2020

Response to Reviewer #2 on the manuscript:

*Using ICESat-2 and Operation IceBridge altimetry for supraglacial lake depth retrievals* by Fair et al.

We thank the reviewer for their comments and suggestions to improve the clarity and structure of the manuscript. In this response, the original comment is given in black, the authors' response in blue, and the proposed changes in orange.

Scope comment: "The paper limits itself to central strong beam (GT2R), but then also includes lower confidence photos from this band which 'decreases measurement precision but gives better agreement with ground-based data.' Because of this, I am wondering why other beams were not used, or at least their potential use discussed in the paper?"

The central strong beam was initially selected for the number of lakes observed over the Amery Ice Shelf. However, we recognize that the other strong beams (GT1L and GT3L) could also be used for depth retrievals. The weak beams (GT1R, GT2R, and GT3R for these ground tracks) are less effective at detecting beds for deeper lakes, so they were omitted from this study. We added the following to Section 2.1 to address these questions:

Our study focused on the central strong beam, as the number of lakes was deemed sufficient for our purposes. While we recognize that the other strong beams could be useful for depth retrievals we did not consider them here. We speculate that the weak beams may avoid issues with multiple scattering and specular reflection, but their power is too low to reliably detect lakes deeper than 4 m.

Page 6 Line 14: "Do you have any estimate for just how widely applicable these methods will be / how easy it is to get good coverage? I understand you have to put limits on this paper somewhere, for sure, so this is mostly out of curiosity and might be of interest in a discussion/ conclusion?"

We agree that these points would be useful to readers, so we added more information in Section 5.1.

The success of this method for lake depth retrievals is governed by spatial and temporal sampling of the instruments across the lakes when they are full. The methods presented here are most effective when the altimeter passes directly over the deep part of a lake rather than at its edge. This provides a lake depth profile that is more representative of the complete lake, allowing for improved estimates of lake depth and extent. A complete lake profile also provides sufficient information to the LSBS algorithm, reducing the risk of false negatives that occur with small lakes or incomplete profiles. The temporal sampling of ICESat-2 and ATM is infrequent (every 91-days for ICESat-2 and random for ATM), and so the same lakes will not always be

present every time these data are required. Therefore, coincident satellite imagery is desirable to simplify the lake-finding process.

Data & Code Citation/Sharing (1) "The Cryosphere's data policy states that 'Authors are required to provide a statement on how their underlying research data can be accessed. This must be placed as the section 'Data availability' at the end of the manuscript.' I did not see such a section. Clarity in citing the exact subsets of the large datasets that you cite would be ideal (which I know is also in your Table 1, but not presented in one place)."

The full information for the ICESat-2 and ATM data is given in Section 3.4, including date, ground track number, and coordinates. However, we acknowledge the lack of a Data Availability section, and we included one in the revised manuscript.

Code and data availability: ICESat-2 ATL03 V002 and ATM L1B V002 data may be accessed from https://doi.org/10.5067/ATLAS/ATL03.002 and https://doi.org/10.5067/19SIM5TXKPGT, respectively. Depth data for the supraglacial lakes given in Figure 4 are available at https://doi.org/10.5281/zenodo.3838274. Depth data for lakes in Figure 3 are available upon request from Zachary Fair. The LSBS algorithm and its subroutines may also be accessed from https://doi.org/10.5281/zenodo.3838274.

Data & Code Citation/Sharing (2) "The Cryosphere guidelines also state that "Data do not comprise the only information which is important in the context of reproducibility. Therefore, Copernicus Publications encourages authors to also deposit software, algorithms, model code, video supplements, video abstracts, International Geo Sample Numbers, and other underlying material on suitable FAIR-aligned repositories/archives whenever possible. These materials should be referenced in the article and cited via a persistent identifier such as a DOI." There is clearly a lot of important code developed and used by the authors, and it would be in line with this journal's goals that it be documented, shared, and cited. This would allow for reproducibility, further application of these methods, and further refinement, as well. I very much hope that the reviewers document, share, and cite the final version of their code to make their methods as open as the data they use and the publication they have chosen to publish in. (Of course, if you have another code/methods paper in prep, then please do cite that and I apologize for jumping the gun!)"

The lake detection algorithms are available under the Assets section through the following link: https://doi.org/10.5281/zenodo.3838274. This is noted in the "*Code and data availability*" section.

The LSBS algorithm and its subroutines may also be accessed from the DOI given above.

Data & Code Citation/Sharing (3) "And since I'm talking about data and code sharing - at the risk of inviting my own citation - you cited Pope at al 2016 on Page 2 Line 23/24. I wonder whether you might (also) want to cite Pope (2016), which I bring up here because if more fully describes, documents, and shares the code developed and used in the Pope et al paper. https://doi.org/10.1002/2015EA000125"

We agree that the given paper would be a useful citation. We therefore added the following to Page 2, Line 17:

The normalized water difference index (NWDI) and dynamics thresholding techniques have also been considered for lake detection (Fitzpatrick et al., 2014; Liang et al., 2012; Moussavi et al., 2016; Pope, 2016; Williamson et al., 2017; Moussavi et al., 2020).

Page 1 Line 8-9 (Abstract): "Can you quickly mention where the uncertainties are derived from here? It might just be me, but if quickly reading, it makes it sounds like there is comparison to some in situ data…"

The uncertainty was derived from the standard deviation of acceptable lake bed photons. In other terms, the depth uncertainty is equal to the spread of lake bed photons. We reworded the given lines to be clearer:

Lake bed uncertainties for these retrievals…

Page 4 Line 10: "about how long is each data granule, in ground distance, to include $10^4$ - $10^5$ photos per window? I think this will help people understand the next assumptions."

Each flight track for ATM is 13-15 km in length, whereas each ICESat-2 ground track is ~$10^3$ km in total distance. We indirectly addressed this comment in response to Reviewer #1:

We divided each data granule into discrete along-track windows to reduce the data volume to ~$10^4$-$10^5$ photons per window. This photon count is equivalent to ~1-10 km in along-track distance for ICESat-2 and ~0.15-1.5 km for ATM.

Page 4 line 14: "How were these ranges selected / chosen? This would seem to be an important part of method development."

These thresholds were selected by comparing the flatness of lake surfaces to that of surrounding ice topography. We also note here that the ATM threshold of 0.002 m was a typo, and it is supposed to be 0.02 m. We added the following to provide more clarity:

We define a "flat" surface for regions where $\sigma \leq 0.05$ m for ATL03 data, and $\leq 0.02$ m for ILATM1B data. We selected these values by comparing the "flatness" of lake surfaces to that of surrounding ice topography.

Page 4 Line 14: "I'm sorry if I missed it, but can you define sigma in the text upon first usage?"

Sigma is previously defined as the standard deviation of high-confidence photons in Lines 12-13. To avoid confusion for future readers, we edited these lines slightly:

We check the flatness of the window by computing the standard deviation ($\sigma$) of high-confidence signal photons…

Page 4 Line 17: "Consider replacing "lake surface" with 'height of the lake surface' and underlining the letters h, s, f, and, c in order to make the abbreviation very clear?"

We changed as requested:

…we assigned the data to a new array for the height of the lake surface ($h_{sfc}$).

Page 4 Line 20: "How were these ranges selected / chosen? This would seem to be an important part of method development."

The ranges for acceptable bed photons were selected through trial-and-error. The given bounds were selected to minimize the impacts of multiple scattering and specular reflection. To make this clear in the text, and to address comments from Reviewer #1, we reworded Lines 19-22 to be:

Within these horizontal bounds, we defined photons as a lake bottom if they satisfied the condition: $h_{sfc} - a\sigma_{sfc} \leq h \leq h_{sfc} - b\sigma_{sfc}$, where $\sigma_{sfc}$ is the standard deviation of lake surface photons. The constraints $a$ and $b$ were derived through trial-and-error, such that $a = 1.0$ (1.8) and $b = 0.5$ (0.75) for ICESat-2 (ATM). We set these constraints to reduce the impacts of multiple scattering and specular reflection on depth estimates.

Page 4 Line 22: "Consider replacing 'lake surface' with 'height of the lake bottom' and underlining the letters h, b, t, and, m in order to make the abbreviation very clear?"

We assume this is a typo and that the reviewer is actually requesting a rewording of the term "lake bottom" (rather than "lake surface"). With this assumption, we applied the following change:

…the data were placed in an array for the height of the lake bottom ($h_{btm}$).

Page 4 Line 27: "How were these filters chosen? This would seem to be an important part of method development."

We expand upon these issues in Section 5.2. However, we acknowledge that justification is needed here, so we applied the following change:

For ICESat-2, lakes shallower than 1.3 m or less than 200 m in horizontal extent were found to be too noisy or ill-defined for further analysis (see Section 5.2 for more details).

Page 4 / Section 3.1 in general: "It would be even clearer to present these methods if there were agreement between the steps here and in Figure 2 (e.g. one box / arrow per bullet point)."

We contemplated this suggestion, and we decided that one box or arrow per bullet point was unnecessary. However, we added labels to Figure 2 (now Figure 1 in response to another suggestion) to improve consistency with Section 3. The modified figure is shown below, and the bullet points in Section 3.1 were changed for consistency:

i. We divided each data granule into discrete along-track windows…

ii. Each data window was binned into elevation-based histograms…

iii. If the satellite image(s) confirmed the presence of a lake, the data were assigned to a new array for the lake surface ($h_{sfc}$). The horizontal extent of the lake surface served as a constraint for where the lake bottom data could be defined…

iv. A series of filters were applied to improve surface/bed estimates…

v. If the data were obtained from ICESat-2, then we followed a photon refinement routine that is described in more detail in Section 3.2. Calculations for lake depth were then performed for both ATM and ICESat-2 retrievals and corrected for refraction (Section 3.3).

Page 6 Line 18: "I could be wrong, but it is possible that Figure 1 and 2 are cited in backwards order? You might consider flipping their numbers?"

You are correct. For better consistency, we changed the numbering for the relevant figures.

(From Page 4, Line 7) For both instruments, the procedure for separation was identical, and is as follows (see Fig. 1 for a schematic view)…

(From Page 6, Lines 16-18) We present cases over the Amery Ice Shelf […], the western Greenland ablation zone […], and Hiawatha Glacier […] (Fig. 2).

Page 5 Line 27: "You mention a refraction correction but then there is no further detail. I know it is pretty basic, but for full clarity perhaps describe slightly more / provide a citation for the method you use for refraction correction?"

The refraction correction mentioned in Line 27 is briefly described in Lines 23-25. Line 24 includes a citation to Parrish et al. (2019), who outline a refraction correction algorithm based on beam angle and water depth. For quick reference, the passage is given below:

As a final adjustment to the lake photons, we applied a refraction correction algorithm to account for slowing down of the light as it enters water. The correction follows the methods utilized by Parrish et al. (2019) by approximating refractive biases as a function of depth and beam elevation angle.

Page 6 Line 29: "I wonder if you this it is important to re-emphasize the filtering of which lake depths were kept in presenting average lake depths? e.g. lots of shallow lakes aren't being included?"

The lake statistics given in Table 2 reflect quantities calculated from the lakes included in Figure 3. Therefore, the excluded lakes were not considered for the average depth given in Line 29. To emphasize this, we reworded Line 29 to be:

The average lake depth estimate for the lakes in Fig. 3 was 1.95 m…

Figure 1: "Consider using dots to indicate location, rather than ovals, which are much larger than the image are?"

This is a good suggestion. We replaced ovals with stars centered on the regions of interest. The stars appeared small if the full continental image was used, so we cropped the images to center on the markers. The new figure may be seen below:

[Figure]

Figure 2: "It is slightly confusing that you use the same blue boxes for both data and processes (e.g. Landsat 8 imagery vs verify lake detection), consider using different shapes / colors / some design choice to indicate the difference?"

We changed the color of the "ATL03/ILATM1B granule" and "Landsat-8 imagery" boxes to green to differentiate data inputs from algorithm steps. We also grouped the steps into sections to create better consistency between the figure and Section 3, as suggested above. The new figure and caption are shown below:

[Figure]

Figure 2. Schematic for the workflow of the lake surface-bed separation algorithm, where green boxes indicate data inputs and blue boxes are steps in the algorithm. Roman numerals match the steps given in Section 3.1.

Table 2: "Please also define d_s, d_p, and L in the table caption"

We changed as requested:

Cumulative statistics for ATM supraglacial lakes explored in this study, including mean and maximum signal-based depth ($d_s$) and polynomial-based depth ($d_p$), along-track extent $L$, mean lake depth uncertainty…

---

## Referee Report (RR1)

**Overview**

Fair et al. (2020) is one of the first studies to incorporate laser altimetry measurements from ICESat-2 and Operation IceBridge to map supraglacial lakes on the Greenland and Antarctic ice sheets. The work presented by the authors falls within the scope of *The Cryosphere* and should make an interesting contribution to a developing field of measuring supraglacial lakes with remote sensing data. The changes made by the authors is substantial and improves both the readability and quality of the manuscript. Overall, this is a promising methods study of supraglacial lakes with laser altimetry measurements, and I recommend publishing the paper as is. I thank the authors for their hard work and I look forward to seeing more from this research in the future.

**References**

Z. Fair, M. Flanner, K. M. Brunt, H. A. Fricker, and A. S. Gardner. Using ICESat-2 and Operation Ice-Bridge altimetry for supraglacial lake depth retrievals. *The Cryosphere Discussions*, 2020:1–21, 2020. doi: 10.5194/tc-2020-136.